# Trends in malaria prevalence and risk factors associated with the disease in Nkongho-mbeng; a typical rural setting in the equatorial rainforest of the South West Region of Cameroon

**Raymond Babila Nyasa**[1,2], **Esendege Luke Fotabe**[1], **Roland N. Ndip**[1,3]*

**1** Department of Microbiology and Parasitology, University of Buea, Buea, Cameroon, **2** Faculty of Science, Biotechnology Unit, University of Buea, Buea, South West Region, Cameroon, **3** Laboratory for Emerging Infectious Diseases, University of Buea, Buea, South West Region, Cameroon

\* ndip3@yahoo.com, ndip.roland@ubuea.cm

**Data Availability Statement:** All relevant data are within the manuscript.

## Abstract

Globally, malaria in recent years has witnessed a decline in the number of cases and death, though the most recent world malaria report shows a slight decrease in the number of cases in 2018 compared to 2017 and, increase in 2017 compared to 2016. Africa remains the region with the greatest burden of the disease. Cameroon is among the countries with a very high burden of malaria, with the coastal and forest regions carrying the highest burden of the disease. Nkongho-mbeng is a typical rural setting in the equatorial rain forest region of Cameroon, with no existing knowledge of the epidemiology of malaria in this locality. This study aimed at determining the current status of malaria epidemiology in Nkongho-mbeng. A cross-sectional survey was conducted, during which blood samples were collected from 500 participants and examined by microscopy. Risk factors such as, age, sex, duration of stay in the locality, housing type, environmental sanitation and intervention strategies including use of, LLINs and drugs were investigated. Trends in malaria morbidity were also determined. Of the 500 samples studied, 60 were positive, giving an overall prevalence of 12.0% with the prevalence of asymptomatic infection (10.8%), more than quadruple the prevalence of symptomatic infections (1.2%) and, fever burden not due to malaria was 1.4%. The GMPD was 6,869.17 parasites/μL of blood (95% C.I: 4,977.26/μL– 9,480.19/μL). A LLINs coverage of 84.4% and 77.88% usage was observed. Unexpectedly, the prevalence of malaria was higher among those sleeping under LLINs (12.56%) than those not sleeping under LLINs (8.97%), though the difference was not significant (p = 0.371). Being a male (p = 0.044), being unemployed (p = 0.025) and, living in Mbetta (p = 0.013) or Lekwe (p = 0.022) and the presence bushes around homes (p = 0.002) were significant risk factors associated with malaria infection. Trends in proportion demonstrated that, the prevalence of malaria amongst patients receiving treatment in the health center from 2015 to 2019 decreased significantly (p < 0.001) and linearly from 9.74% to 3.08% respectively. Data

**Funding:** The authors received no specific funding for this work.

**Competing interests:** The authors have declared that no competing interest exist.

generated from this study can be exploited for development of a more effective control measures to curb the spread of malaria within Nkongho-mbeng.

## Introduction

Malaria is the most prevalent mosquito-borne parasitic disease throughout tropical and subtropical regions of the world. An estimated 228 million cases of malaria occurred worldwide in 2018, accounting for nearly 405000 deaths from malaria globally, compared to 416 000 estimated deaths in 2017, and 585000 in 2010, of which 94% were in WHO African region [1]. However, Cameroon remains among 11 countries that account for 92% of the malaria infection in sub-Saharan-Africa [2]. The disease is endemic in Cameroon with the level of endemicity varying from one ecological zone to another. Studies carried out to investigate the epidemiology of malaria in the South West region of Cameroon have been limited to the slope of Mount Cameroon in Fako Division. However, Nkongho-mbeng in the Nguti Sub-Division defers from Fako division, with its pristine forest vegetation consisting of vast surface areas of unexploited timber and numerous valleys and hills. The poor earth road network and absence of industries and higher institutions of learning, makes the standard of living significantly lower than that of inhabitants of Fako division. Consequently, the epidemiology of malaria may vary in this setting compared to what is seen on the slope of Mount Cameroon.

The Ministry of Public Health in Cameroon reported that malaria causes 30 to 35% of overall mortality and 67% of annual childhood mortality [3]. Insecticide treated nets (ITNs) and indoor residual spraying (IRS) have both been demonstrated to reduce malaria [4] and, to date, are the methods mainly used for controlling malaria vectors and associated malaria transmission [5]. Nevertheless, long lasting insecticide-treated bed nets (LLINs) are the major and most promising components of the selective vector control strategies [1], and it is the major malaria control measure used by the Cameroon Ministry of Public Health to fight against the disease and its vector. The poor road network of Nkongho-mbeng may have hindered the level of coverage in the distribution of LLINS. It has been shown in Fako division of the South West Region of Cameroon that LLINS ownership was lower in rural settings than in Semi-Urban communities [6], and, a high malaria prevalence of 30.7% was reported during the rainy season, in Bolifamba following the introduction of LLINs [7]. So far, no study has evaluated the coverage of LLINS in Nkongho-mbeng as well as the level of endemicity of malaria parasite in this community. This study, therefore, aimed at determining the trend and risk factors of malaria within these communities.

Nkongho-mbeng, is an example of a typical rural setting in the equatorial rain forest region of Cameroon. Most of its inhabitants are farmers with low level of education, mainly practicing subsistence farming. Thus, it is probable that the inhabitants may have little or no knowledge about the malaria parasite and the appropriate prevention strategies. This would probably lead to a high burden of the disease when coupled with the enabling environment of the equatorial rain forest of varied topography (highlands, slopes and lowland areas) and meandering streams within settlements, serving as breeding grounds for the malaria vector.

## Materials and methods

### Study area

This study was carried out in Nkongho-mbeng, a clan in the Nguti sub-division of the Kupe-Manenguba division of the South West Region of Cameroon. Nkongho-mbeng is made up of three villages (Dinte, Mbetta and Lekwe), making up the Mbetta health area of the Fontem

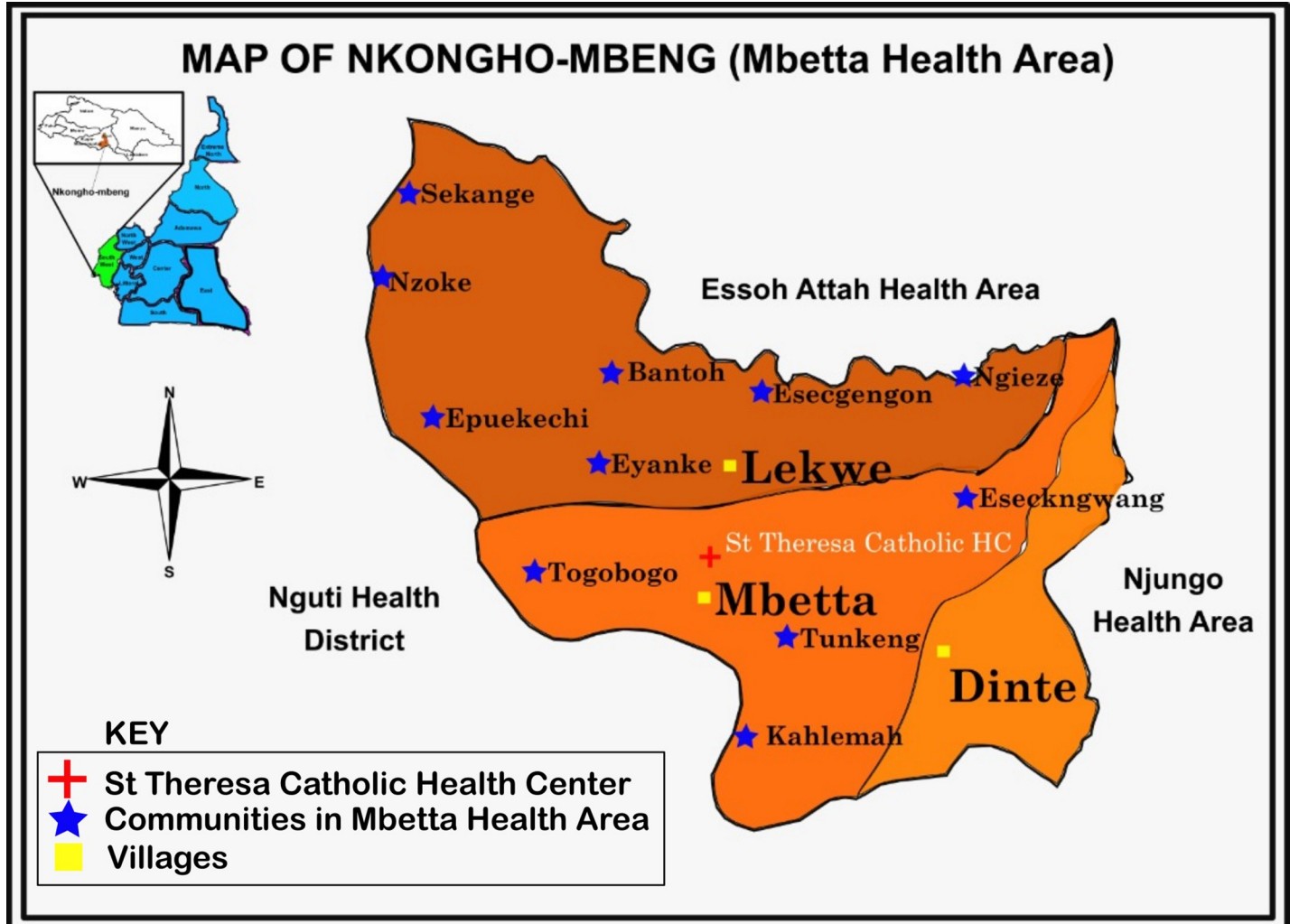

**Fig 1. Map of Nkongho-mbeng (Mbetta health area) in the South West Region of Cameroon.**

health district Fig 1. It is located between latitude $5^0$19' and $5^0$22'E and longitude $9^0$40' and $9^0$50'N. The topography is slightly sloppy, surrounded by hills; an earth road, cuts across the three villages and one fast flowing river (the Mekoh river) flows across the three villages. Many other smaller streams are located at different places within the village. Nkongho-mbeng has a typical rain forest vegetation with almost every compound having crops like plantain, banana, cocoa and coffee around their home. The locality experiences two major seasons, dry and rainy seasons. The dry season typically runs from November to February, while the rainy season starts in March and ends in October. The walls of the houses in Nkongho-mbeng are made of mud block and a few with cement block. Majority of the inhabitants are famers and depend solely on agriculture for food and income.

## Study design

The study was carried out in two arms. The first arm was a cross-sectional community based household survey conducted in the month of July 2019, during which participants were met at

their homes in the evening, when most people were back from their farms and were interview face-to-face or his/her guardian, in the case of under age. Blood samples for microscopy were collected and taken to the St. Theresa Catholic Health Center Mbetta laboratory for analysis. The second arm of the study involved a retrospective collection of data from the South West Regional Delegation of Public Health, Cameroon; during which data of the hospital prevalence from 2015 to 2019 was collected and used for determination of trends in malaria prevalence in Nkongho-mbeng.

## Sample size calculation

The minimum sample size was calculated using the Swinscow Formula (Swinscow *et al.*, 2002).

$$n = \frac{Z^2 P(1-P)}{d^2}$$

Where; n = minimum sample size for this study;
Z = confidence level,
P = prevalence from a similar study
d = level of precision
(d = 0.05, Z = 1.96, P = 0.5)

$$\mathbf{n} == \frac{(1.96)^2(0.5)(1-0.5)}{(0.05)^2} \, 385 \text{ participants required}$$

A total of 500 participants i.e males and females of all age groups who gave their consent and, ascent in case of minors, were sampled. Those who did not give their consent or assent, people who were not resident in Nkongho-mbeng and, people who had taken malaria drugs two weeks before the survey were not allowed to take part in this study.

## Ethical considerations

Ethical clearance for this study was obtained from the University of Buea Faculty of Health Sciences, Institutional Review Board (Ref: 2019/977-05/UB/SG/IRB/FHS). Administrative clearance was obtained from the Ministry of Public Health, Regional Delegation for the South West Region (Ref: R11/MINSANTE/SWR/RDPH/PS/887/652). Further authorization was obtained from the Fontem Health District Service (Ref: 06MINSANTE/RDSW/DHSF/Vol2/013), chiefs and quarter heads of the Nkongho-mbeng villages. Participants also signed an informed consent form before taking part in the study. For participants who were unable to read or write, the information was read and explained to them and their tomb prints were taken.

## Questionnaire administration

A semi-structured questionnaire developed for the purpose of this study was used (supplementary 1). It was pretested in Muea, a quarter in Buea, under Fako division, before being administered to the participants, to collect data on demographics (sex, age, literacy, occupation and marital status); Socioeconomic status (number of house occupants, house type, toilet type, nature of house surroundings, water sources and storage) and, malaria treatment and prevention practices (long lasting insecticide nets (LLINs) ownership, and usage) and indoor residual spray (IRS) usage. Use of LLIN was defined as sleeping under a LLIN every night prior to the survey.

## Sample collection

Body temperature of the participants who gave their consent were measured using a thermometer and a finger prick was done using a sterile disposable lancet, to obtain blood sample for microscopic analysis. Data for trends in hospital prevalence of malaria in Nkongho-mbeng was collected from the malaria unit at the South West Regional Delegation of Public Health, Cameroon.

## Laboratory analysis

Thick blood films made from participants' samples, were air-dried and transported to the St Theresa Catholic Health Center Mbetta, where they were stained with 5% Giemsa for fifteen to thirty minutes, rinsed, air dried and observed under the light microscope at X100 objective (oil immersion). A thick blood smear was declared negative, after observing more than 100 fields at X100 high power magnification, and no malaria parasite seen. Positive slides were quantified by counting the number of parasites against 200 white blood cells, and the parasites/µl of blood was calculated by assuming a leucocyte count of 8000 per microliter as reported elsewhere [8].

## Statistical analyses

Data collected was analyzed using STATA, IBM-Statistical Package for Social Sciences (IBM-SPSS) version 25 and Epi-info version 7. Prevalence was calculated by dividing the number of positive samples by the total sample size. The data was tested for normality using the Kolmogorov-Smirnov (K-S) test and the data was not normally distributed, therefore, Kruskal-Wallis test and Mann-Whitney tests were used for analysis of continuous variables (parasite load). Chi-square analysis and Logistic regression were also used for analysis of categorical variables and determination of risk factors was by bivariate and multivariate analysis. Multicollinearity analysis was carried out, to determine possible risk factors which could colinearly influence the regression model. The Chi-squared test for trends in proportion was used to determine trends in malaria prevalence over time. Graphical presentation of the data was done using Microsoft Excel 2013, and SPSS version 25.

## Results

### Prevalence of malaria in Nkongho-mbeng

Of the 500 blood samples examined, 60 were positive for malaria infection using microscopy giving an overall prevalence of malaria in Nkongho-mbeng of 12.0%, with the prevalence of asymptomatic malaria infection of 10.8% (54/500), which more than quadruples the prevalence of symptomatic malaria infection of 1.2% (6/500) Fig 2. The percentage of fever observed in the community, which was not caused by malaria was 1.4% (7/500). Most (86.6%) of the participants were apparently healthy.

### Prevalence of malaria in relation to demographic factors

The prevalence of malaria was significantly associated with age (p < 0.001), gender (*p* = 0.042), location of villages (p = 0.022) and occupation (p < 0.001) as shown on Table 1. The prevalence of malaria was higher among males (14.90%) than females (8.98%), though the difference was marginally significant (p = 0.042). Malaria infection was most common among infants (26.09%) than any other age category. Equally, malaria prevalence was highest among the unemployed (25.49%) and the inhabitants of Mbetta (14.29%).

   Logistic regression analysis was carried out between malaria prevalence as dependent variable and socio-demographic as independent variables, for all the variables which were significant by bivariate analysis. In the bivariate analysis, males (cOR = 1.78, *p* = .044), being

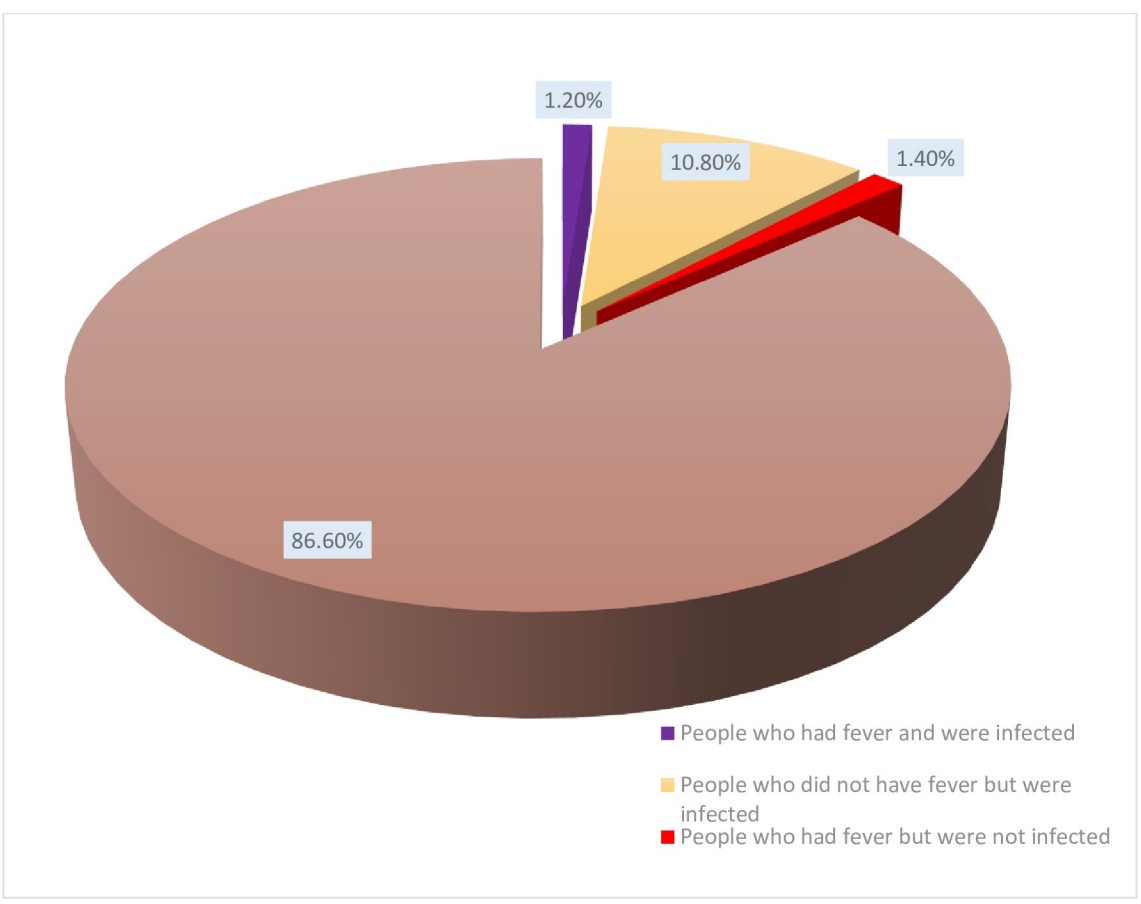

**Fig 2. Prevalence of malaria in Nkongho-mbeng.**

unemployed (cOR = 10.95, $p$ = .025), those residing in Lekwe (cOR = 9.04, $p$ = 0.032) and Mbetta (cOR = 10.33, $p$ = 0.023) were significantly associated with a higher prevalence of Malaria infection among inhabitants of Nkongho-mbeng. Nonetheless, it was observed that increase in age ($p$ < 0.001) and increase in income level ($p$ = .017) significantly reduced the odds of being malaria infected. Hence, age (cOR = 0.15) and income (cOR = 0.42) levels were considered as protective factors of malaria infection. After performing the multivariate logistic regression analysis, being unemployed ($p$ = .015), being a pupil ($p$ = 0.027), as well as living in Lekwe ($p$ = .022) and Mbetta ($p$ = 0.013) were found to be the significant risk factors of malaria among the people of Nkongho-mbeng.

Multicollinearity analysis revealed that all the variables associated with income levels had variance inflation factor less than 3 (Table 2). So, there is no evidence of multicollinearity amongst the risk factors associated with income level.

## Environmental and behavioral characteristics of the study population in relation to malaria prevalence

Participants living in bushy areas had significantly ($p$ < 0.001) higher malaria prevalence (23.23%) than those living in non-bushy areas (9.23%) Table 3. Similarly, those residing around stagnant water had significantly ($p$ = 0.045) higher malaria prevalence (15.35%) than

**Table 1. Prevalence of malaria in relation to demographic factors (N = 500).**

| Characteristic | Category | Frequency (%) | Prevalence (%) | p-value chi square | Bivariate analysis COR (95% CI) | P value | Multivariate analysis AOR (95% CI) | P value |
|---|---|---|---|---|---|---|---|---|
| **Gender** | Female | 245(49.0) | 22 (8.98) | | 1 | | 1 | |
| | Male | 255 (51.0) | 38 (14.90) | | 1.78 (1.02–3.10) | | 1.91 (1.06–3.47) | .032 |
| | **Sex ratio (M/F)** | **1.15** | $\chi^2$ = 4.150 | **0.042*** | | **.044*** | | |
| **Age Group (years)** | | | | | 1 | | **1** | |
| | Infants (0<2) | 23 (4.6) | 6 (26.09) | | 0.52 (0.23–1.18) | .117 | 0.84 (.26–2.73) | .778 |
| | Children (2–10) | 135(27.0) | 22 (16.29) | | 0.73 (0.35–1.51) | .388 | 2.56 (.71–9.28) | .151 |
| | Teenagers (>10–18) | 89 (17.8) | 19 (21.35) | | | | | |
| | Adults (>18–100) | 253(50.6) | 13 (5.38) $\chi^2$ = 25.327 | **<0.001*** | 0.15 (0.06 –.41) | **< .001*** | 1.10 (.13–9.70) | .929 |
| **Mean ± S.D age** | **25.30 ± 21.30** | | | | | | | |
| **Village** | Dinte | 63 (12.6) | 1 (1.59) | | 1 | | **1** | |
| | Lekwe | 220 (44.0) | 28 (12.73) | | 9.04 (1.21–67.83) | **.032*** | 10.83(1.41–83.14) | **.022** |
| | Mbetta | 217(43.4) | 31 (14.29) $\chi2$ = 7.652 | **0.022*** | 10.33 (1.38–77.27) | **.023*** | 13.17 (1.71–101.1) | **.013** |
| **Educational level** | No school | 85 (17.0) | - | | 1 | | - | - |
| | Nursery | 17 (3.4) | | | 1.09 (0.28–4.29) | .905 | - | - |
| | Primary | 254(50.8) | | | 0.71 (0.36–1.40) | .317 | - | - |
| | Secondary | 120 (24.0) | | | 0.46 (0.19–1.10) | .079 | - | - |
| | High school | 20 (4.0) | | | 0.56 (0.12–2.71) | .474 | - | - |
| | University | 4 (0.8) | | | 0 | .999 | - | - |
| **Occupation** | Business | 33 (6.6) | 1 (3.03) | **< 0.001*** | 1 | | 1 | |
| | Farmer | 158 (31.6) | 7 (4.43) | | 1.48 (0.18–12.48) | 0.717 | 1.04 (.11–9.94) | 0.971 |
| | Health worker | 11 (2.2) | 0 (0.0) | | 0.00 | 0.999 | 0.00 | 0.999 |
| | Pupil | 143 (28.6) | 27 (18.88) | | 7.45 (0.97–56.94) | 0.053 | 71.64 (1.61–3185) | **0.027** |
| | Student | 88 (17.6) | 10 (11.36) | | 4.10 (0.50–33.38) | 0.187 | 21.94 (0.54–900.0) | 0.103 |
| | Teacher | 16 (3.2) | 2 (12.50) | | 4.57 (0.38–54.66) | 0.230 | 4.76 (0.35–65.30) | 0.243 |
| | Unemployed | 51 (10.2) | 13 (25.49) $\chi2$ = 27.826 | | 10.95 (1.36–88.30) | **0.025*** | 139.27 (2.62–7401) | **0.015** |
| **Marital status** | Single | 341(68.2) | - | - | | | | |
| | Married | 134 (26.8) | | | | | | |
| | Widow | 24 (4.8) | | | | | | |
| | Widower | 1 (0.2) | | | | | | |
| **Religion** | Christian | 478 (95.6) | - | - | | | | |
| | Muslim | 6 (1.2) | | | | | | |
| | Pagan | 16 (3.2) | | | | | | |
| **House type** | Cement | 88 (17.6) | - | - | 1 | | - | - |
| | Mud | 412 (82.4) | | | 0.84 (0.42–1.65) | 0.603 | - | - |
| **1Household size** | 1–10 | 387(77.4) | - | - | | | | |
| | > 10 | 113(22.6) | | | | | | |
| **Income level (CFA)** | < 25,000 | 320(64.0) | - | - | 1 | | 1 | |
| | 25,000–50,000 | 142(28.4) | | | 0.42 (0.21–0.85) | **0.017*** | 14.76(.59–369.9) | 0.101 |
| | > 50,000 | 38 (7.6) | | | 0.15 (0.02–1.12) | 0.064 | 3.32 (.12–94.49) | 0.483 |
| **Toilet type** | Pit latrine | 462 (92.4) | - | - | | | | |
| | Flushing toilet | 31(6.2) | | | | | | |
| | No toilet | 7 (1.4) | | | | | | |

S.D = Standard Deviation M/F ratio = Male to Female ratio COR: Crude Odds Ratio AOR: Adjusted Odds Ratio.

* Statistically significant association, p < 0.05 $\chi^2$ = Pearson's Chi square test.

those residing in areas with no stagnant water (9.47%). Though not significant (p = 0.814), malaria prevalence was higher among those living around streams (12.29%) compared to those not living around streams (11.59%).

**Table 2. Variance inflation factor values for multicollinearity of variables associated with income level in Nkongho-mbeng.**

| Independent variable / Dependent variable | House type | House size | ceiling type | window nets | Toilet type | Educational level | monthly income |
|---|---|---|---|---|---|---|---|
| House type | - | 1.647 | 1.488 | 1.778 | 1.674 | 1.825 | 1.786 |
| House size | 1.199 | - | 1.321 | 1.316 | 1.325 | 1.329 | 1.318 |
| Ceiling type | 1.450 | 1.769 | - | 1.482 | 1.776 | 1.777 | 1.776 |
| Window nets | 1.398 | 1.422 | 1.196 | - | 1.316 | 1.435 | 1.437 |
| Toilet type | 1.221 | 1.328 | 1.330 | 1.220 | - | 1.331 | 1.333 |
| Educational level | 1.073 | 1.074 | 1.072 | 1.073 | 1.073 | - | 1.027 |
| monthly income | 1.090 | 1.105 | 1.112 | 1.114 | 1.115 | 1.065 | - |

In the bivariate analysis, those residing around bushy areas (cOR = 2.98, p < .001) and areas of stagnant water (cOR = 1.73, p = 0.047) were significantly associated with malaria infection. After performing the multivariate logistic regression analysis, only living around bushy areas (p = 0.002), was found to be a significant environmental risk factors for malaria infection in Nkongho-mbeng.

**Table 3. Environmental and behavioral characteristics of the study population in relation to malaria prevalence.**

| Variable | Category | Frequency | Percentage | prevalence | P value Chi square | Bivariate analysis COR (95% CI) | P value | Multivariate analysis AOR (95% CI) | P value |
|---|---|---|---|---|---|---|---|---|---|
| **Use of insecticides spray** | Yes | 22 | 4.4 | | | 1 | | | |
| | No | 478 | 95.6 | 60 (12.0) | p = .076 | 0 | 0.998 | – | – |
| **Presence of bushes** | Yes | 99 | 19.8 | 23 (23.23) | | 1 | | 1 | |
| | No | 401 | 80.2 | 37 (9.23) | p < .001* | 2.98(1.67–5.30) | < .001* | 2.72 (1.44–5.13) | **0.002*** |
| **Presence of stream around home** | Yes | 293 | 58.6 | 36 (12.29) | | 1 | | | |
| | No | 207 | 41.4 | 24 (11.59) | p = .814 | 1.07 (.62–1.85) | 0.814 | | |
| **Presence of stagnant water** | Yes | 215 | 56.7 | 33 (15.35) | | 1 | | 1 | |
| | No | 164 | 43.3 | 27 (9.47) | p = .045* | 1.73 (.01–2.98) | **0.047*** | 1.22 (0.67–2.24) | 0.515 |
| **Use of window nets** | Yes | 75 | 15.0 | | | | | | |
| | No | 425 | 85.0 | | | | | | |
| **Ceiling type** | Bamboo | 22 | 4.4 | 4 (18.18) | | 1 | | | |
| | Plywood | 151 | 30.2 | 19 (12.58) | | 1.67 (.60–1.95) | 0.377 | | |
| | Zinc | 12 | 2.4 | 0 (0) | | 1.08 (.54–5.20) | 0.795 | | |
| | No ceiling | 315 | 63.0 | 37 (11.75) | p = .475 | .000 | 0.999 | | |
| **Source of water** | Tap | 93 | 18.6 | 8 (8.60) | | 1 | | | |
| | Stream | 123 | 24.6 | 22 (17.89) | | 1.53 (.34–6.80) | 0.578 | | |
| | Spring | 257 | 51.4 | 28 (10.89) | | 2.72 (.60–12.35) | 0.194 | | |
| | River | 27 | 5.4 | 2 (7.41) | p = 0.117 | 1.18 (.24–5.90) | 0.843 | | |
| **Water storage method** | Both | 424 | 84.8 | 49 (11.56) | | 1 | | | |
| | Close containers | 44 | 8.8 | 6 (13.64) | | 1.21 (.49–3.01) | 0.684 | | |
| | Open containers | 32 | 6.4 | 5 (15.63) | p = .745 | 1.42 (.52–3.85) | 0.494 | | |

COR: Crude Odds Ratio AOR: Adjusted Odds Ratio * Statistically significant association, $p < 0.05$ $\chi^2$ = Pearson's Chi square test.

**Table 4. Prevalence of malaria in relation to malaria management characteristics of participants.**

| Variable | Category | Frequency | Percentage | Prevalence | Significance |
|---|---|---|---|---|---|
| **Have you had malaria before?** | Yes | 409 | 81.8 | 57(11.4) | $X^2$ = 7.980, p = .005* |
| | No | 91 | 18.2 | 3(0.6) | |
| **If yes when last were you sick of malaria?** | one month ago | 63 | 12.6 | 7(1.4) | $X^2$ = 21.782, p < .001* |
| | five months ago | 224 | 44.8 | 39(7.8) | |
| | 1 year and above | 120 | 24.0 | 11(2.2) | |
| **Where do you take treatment?** | Auto medication | 35 | 7.0 | - | $X^2$ = 16.798, p = .002* |
| | drug store | 137 | 27.4 | 18(3.6) | |
| | Health center | 225 | 45.0 | 36(7.2) | |
| | herbalist | 12 | 2.4 | 3(0.6) | |
| **drugs used** | Athermeter | 106 | 21.2 | 14(2.8) | $X^2$ = 10.053, p = .040* |
| | Quinine | 60 | 12.0 | 9(1.8) | |
| | herbs | 39 | 7.3 | 3(0.6) | |
| | don't know | 203 | 40.6 | 31(6.2) | |

## Prevalence of malaria in relation to malaria management characteristics of study participants

The prevalence of malaria was significantly (p = 0.005) higher among those who had been infected with malaria before (11.4%) than those who have not been infected with malaria before (0.6%) Table 4. Those who were lastly diagnose of malaria five months prior to the survey had the highest prevalence (7.8%) compared to those who were diagnoses of malaria one month (1.4%), and one year (2.2%) prior to this study. Majority of the participants (45%) commonly receive their malaria treatment from Health centers, although 7% practice auto-medication, while 2.4% depend on herbalist for treatment. Surprisingly, people who practice auto medication had zero prevalence of malaria. The commonly reported drugs used for treatment of malaria in these communities were Athermeter (21.2%), quinine (12%) and herbs (7.3%).

## Relationship between the prevalence of malaria and duration of stay in Nkongho-mbeng

Chi square test showed a significant relationship between malaria prevalence and duration of stay ($X^2$ = 26.852, p < 0.001). Participants who had resided for 10–15 years in Nkongho-mbeng had the highest prevalence of malaria compared to those in the other groups as indicated on Fig 3.

## Parasite density of malaria

The overall geometric mean parasite density (GMPD) of malaria among the people of Nkongho-mbeng was 6,869.17 parasites/μL of blood (95% C.I: 4,977.26/μL– 9,480.19/μL) Table 5. The GMPD in males (8,467.40 parasites/μL) was significantly (p = 0.030) higher than in females (4,786.16 parasites/μL). Equally, the GMPD of malaria was significantly (p < 0.001) higher among teachers (26,037.78/μL) than those in other professions, (H = 28.172/μL). With regards to marital status, the GMPD was significantly (p = 0.001) higher among singles (7000.12 Parasites/μL) than those married (5795.57 Parasites/μL), (H = 15.420/μL,).

## Parasite density with respect to age groups

The parasite density of malaria was significantly associated with age (H = 23.209, p < 0.001). Adults (835 parasites/μL) had the lowest parasite density of malaria while infants (6712 parasites/μL) had the highest parasite loads of malaria in the study as shown in Fig 4.

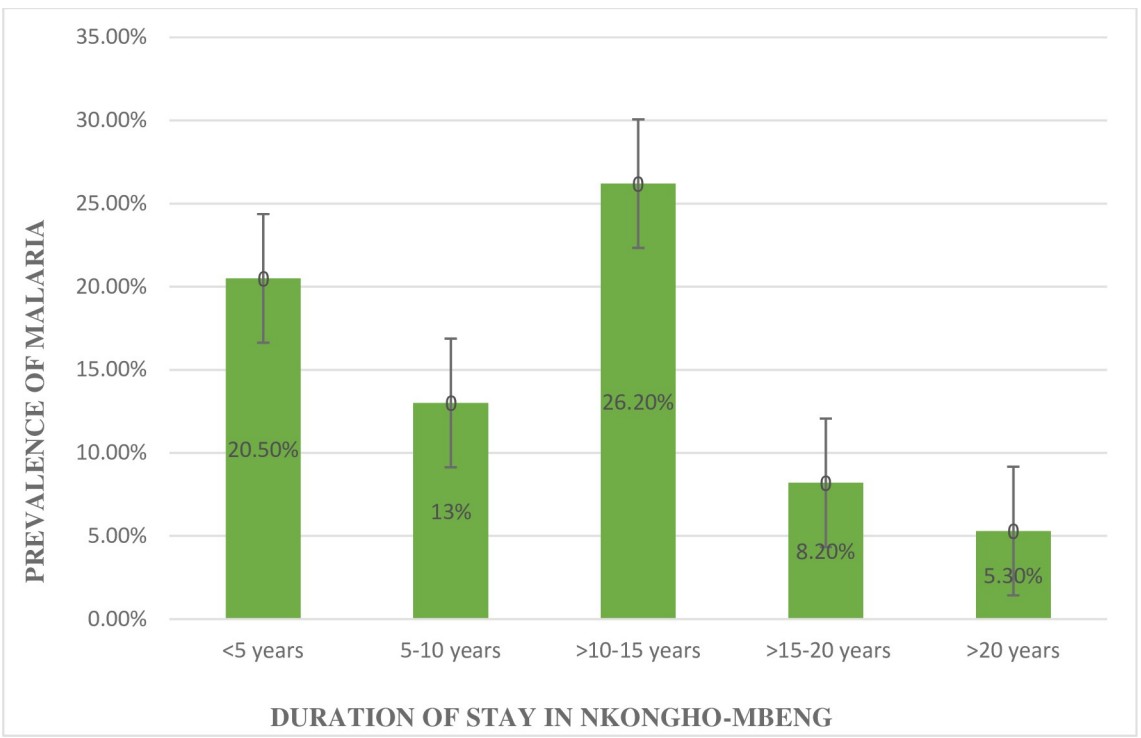

**Fig 3. Relationship between the prevalence of malaria and duration of stay in Nkongho-mbeng.**

## Parasite density with relation to duration of stay in the community

The mean malaria parasite density was significantly associated with duration of stay in the community (H = 27.02, p < 0.001). Participants who had resided in Nkongho-mbeng for less

**Table 5. Geometric mean parasite density of malaria with respect to demographic factors.**

| Characteristics | Category | Number examined | GMPD (Parasites/µL) | Statistics | *P value* |
|---|---|---|---|---|---|
| **Overall GMPD** | | | **6869.17** | | |
| Sex | Female | 245 | 4,786.16 | U = 29255 | **0.030**[*] |
| | Male | 255 | 8,467.40 | | |
| Occupation | Business | 33 | 7,407.41 | H = 28.172 | **< 0.001**[*] |
| | Farmer | 158 | 5,834.80 | | |
| | Health worker | 11 | 0 | | |
| | Pupil | 143 | 6,315.56 | | |
| | Student | 88 | 5,319.37 | | |
| | Teacher | 16 | 26,037.78 | | |
| | Unemployed | 51 | 8,804.98 | | |
| Marital status | Single | 341 | 7000.12 | H = 15.420 | **0.001**[*] |
| | Married | 134 | 5795.57 | | |
| | Widow | 24 | 0 | | |
| | Widower | 1 | 0 | | |

[*] Significant association, *p* < 0.05 H = Kruskal-Wallis test U = Mann-Whitney.

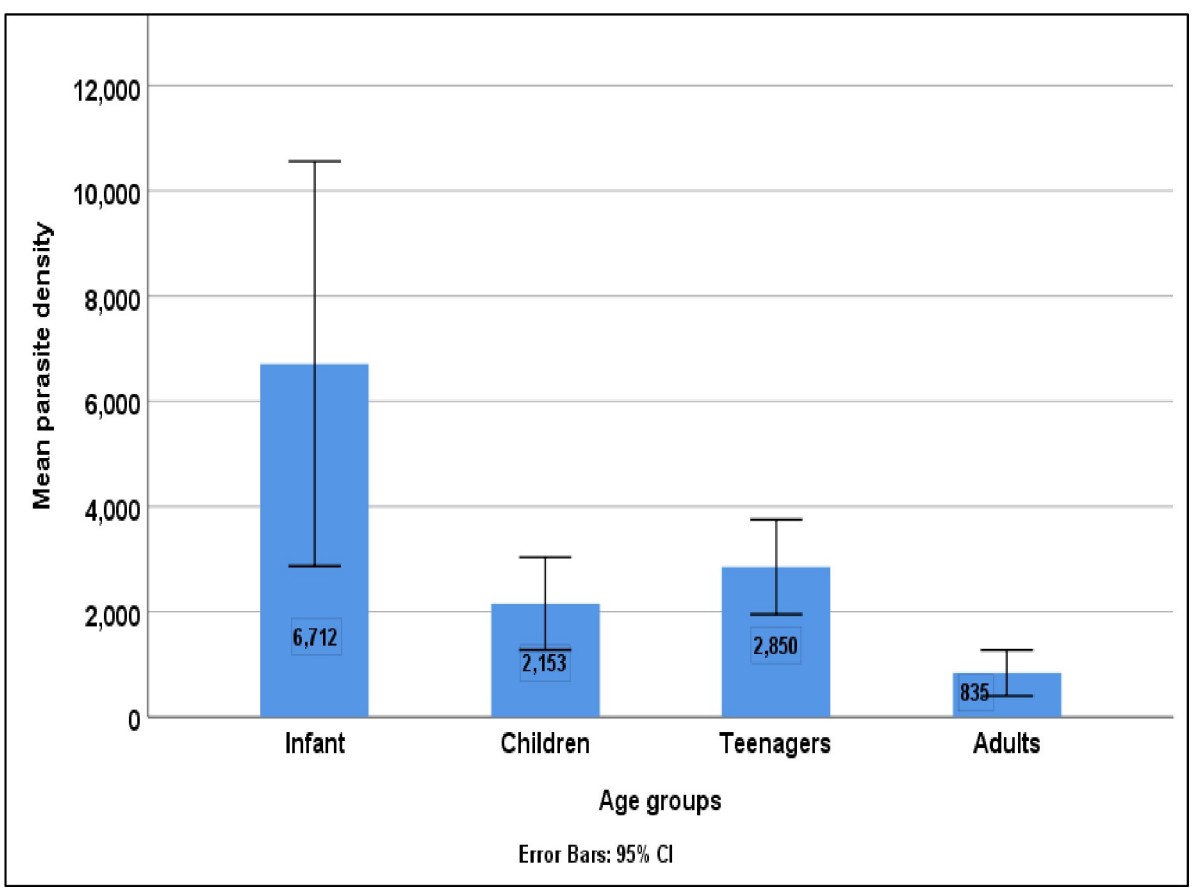

**Fig 4. Parasite density with respect to age groups.**

than 5 years had higher malaria parasite loads (4272.41 parasites/µL) than those who had lived for 5 years and above as illustrated on Fig 5.

## Relationship between mean parasite density of malaria and income level

A Spearman's rank correlation showed that, the mean parasite density of malaria decreased significantly with increase in income level (r = - 0.137, p = 0.002) (Fig 6).

## Parasite density of malaria with respect to the participants' villages

The geometric mean parasite density of malaria was significantly higher among the inhabitants of Dinte village (33600 parasites/µL) compared to those residing in Lekwe (6724 parasites/µL) and Mbetta (6653 parasites/µL); (H = 7.394, *p* = 0.025) as shown in Fig 7.

## Determination of LLINs coverage and usage in Nkongho-Mbeng

Up to 422 of the 500 participants owned LLINs at home giving a percentage LLINs coverage in Nkongho-Mbeng of 84.4% as shown in Fig 8A. Most (70.28%) of the LLINs used in Nkongho-Mbeng were old LLINs ie more than three years old (Fig 8B). Majority (77.88%) of the inhabitants slept under LLINs daily. Some (20.94%) slept infrequently under LLINs while a few (1.18%) did not use LLINs (Fig 8C).

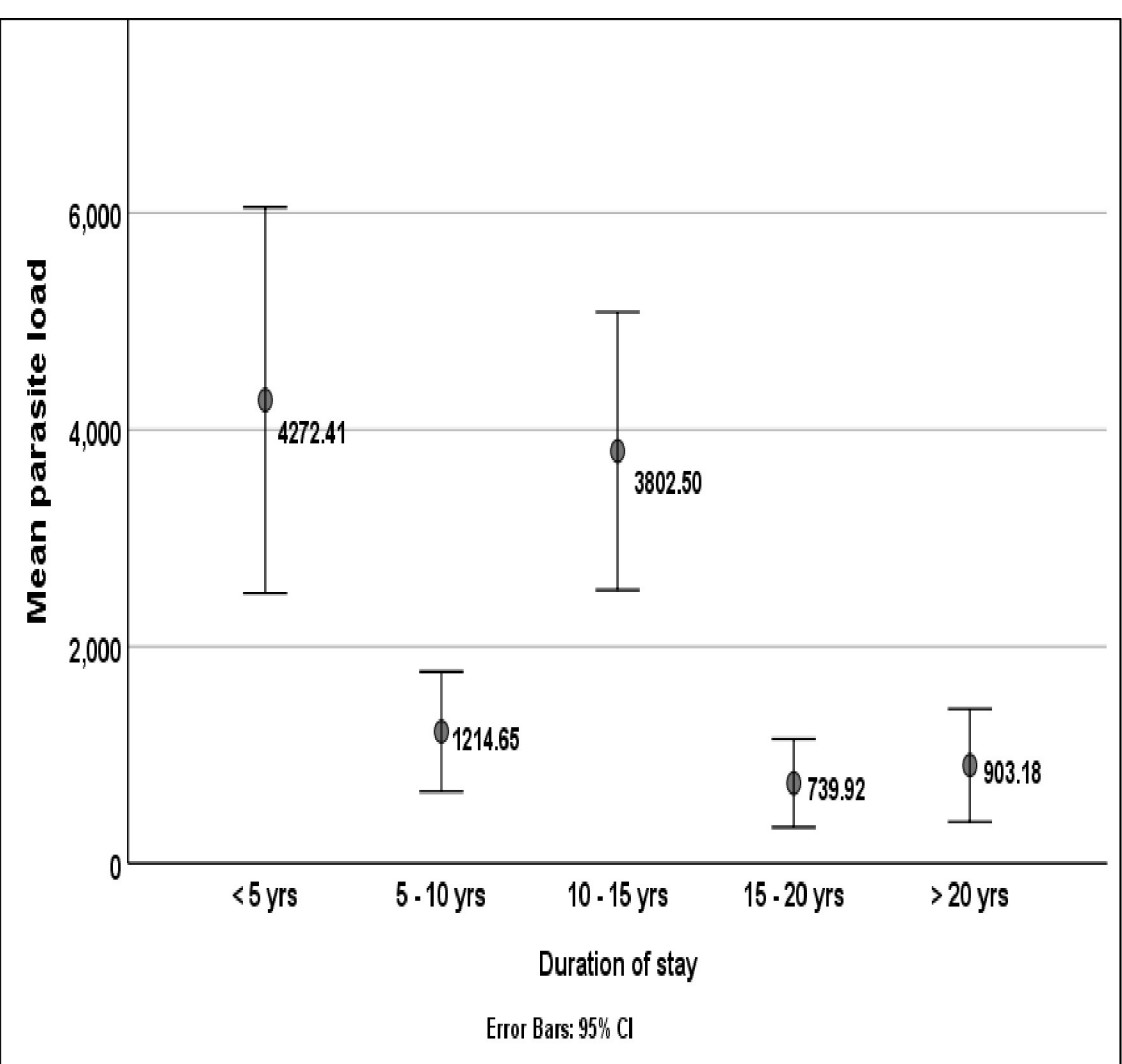

**Fig 5. Parasite density with relation to duration of stay.**

### Prevalence and parasite density of malaria with respect to utilization of LLINs

Unexpectedly, the prevalence of malaria was higher among those sleeping under LLINs (12.56%) than those not sleeping under LLINs (8.97%), though the difference in prevalence was not significant (p = 0.371) Table 6. However, the GMPD of malaria was higher among those not sleeping under LLINs (10659.07/μL) than those sleeping under LLINs (6481.89/μL), although the difference was not significant (p = 0.399). In terms of frequency of LLINs used, those that slept under LLINs infrequently (13.48%) had higher malaria prevalence than those that slept under LLINs daily (12.69%) although the difference was not statistically significant (p = 0.678). The parasite load was similar among those that slept under LLINs daily (6351.64/μL) and those that slept under LLINs infrequently (6007.99/μL, p = 0.684).

### Trends in the prevalence of malaria from 2015–2019 in Nkongho-mbeng

The Chi-squared test for trends in proportion among hospital attending patients demonstrated that, the prevalence of malaria per 1,000 inhabitants decreased significantly and linearly from 9.74% in 2015, to 3.08% in 2019 ($X^2$ = 406.64, p < 0.001) as shown on Fig 9.

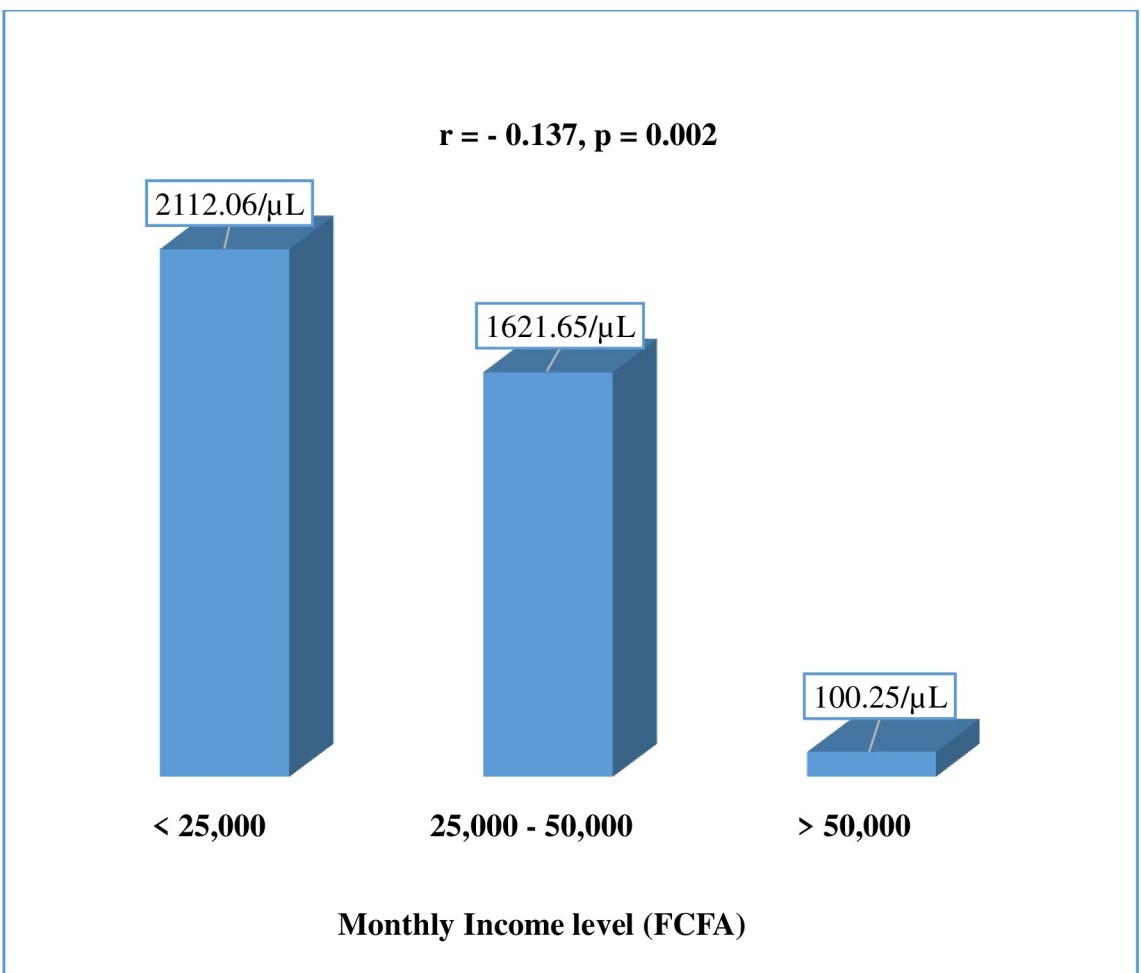

**r = - 0.137, p = 0.002**

**Fig 6. Relationship between mean parasite density of malaria and income level.**

## Discussion

The overall prevalence of malaria parasite infection in Nkongho-mbeng was found to be 12.0%. This observed prevalence is in line with 14.5% and 8.8% prevalence observed in a study in Bamenda and Ngaoundere, respectively, which are towns found in different epidemiological strata within Cameroon [9]. Based on the classification of malaria endemicity [10], Nkongho-mbeng could be ranked on the scale of *Plasmodium falciparum* parasite rate for the age group 2–10 years ($pfpr_{2-10}$) as belonging to a mesoendemic stratum where transmission is high, under normal rainfall conditions and low during the dry season.

However, the overall prevalence of malaria in Nkongho-mbeng in this study (12%) was lower than that reported in other areas of the South West Region of Cameroon by previous studies; 27.7%, 33.3%, 33.7%, 20.1% and 27.4% [7,6,11,12], in the Mount Cameroon area and in Limbe [9], respectively. The low prevalence observed in this study could be as a result of the period, during which this study was carried out (July); the peak of the rainy season with frequent heavy rains with too much run-offs, which might have washed off breeding sites of the mosquitos vectors.

Prevalence of malaria infection was higher among infants (26.09%) than any other age group category. This observation in prevalence, among infants, is in line with other studies

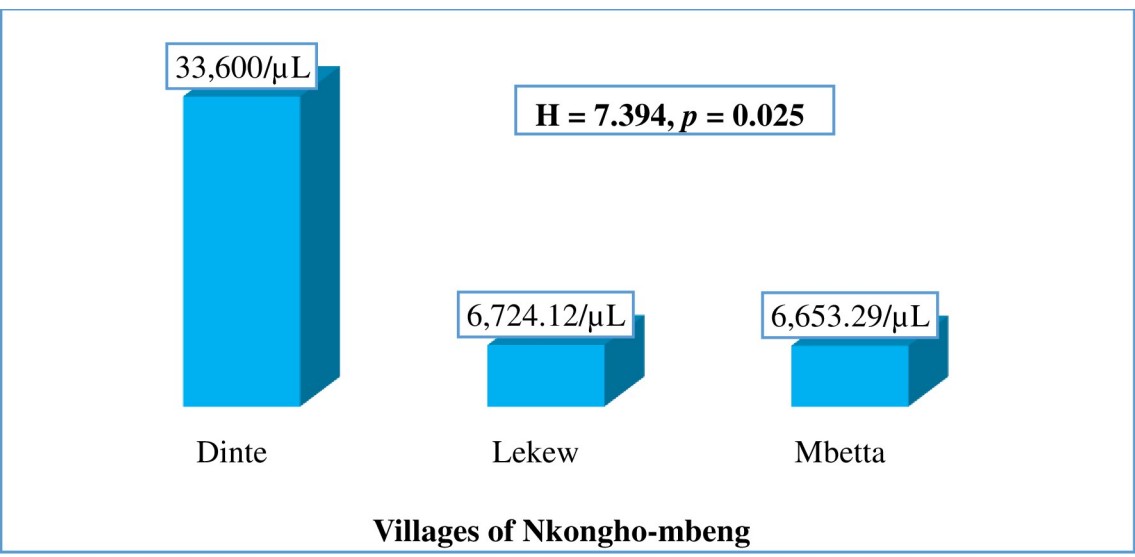

**H = 7.394, *p* = 0.025**

**Fig 7. Geometric mean parasite density of malaria with respect to villages examined.**

[6,11] carried out in some rural and urban settings in the South West Region of Cameroon with a prevalence of 27.7% and 29.6%, respectively in children <15 years of age. In contrast, higher prevalence of 46.7% and 66.9% in children less than 15 years has been reported in other areas in the Mount Cameroon area [13,14].

Malaria prevalence was also observed to be significantly associated with gender (*p* = 0.042), with males having a higher prevalence (14.90%) compared to females (8.98%), though the difference was minimal. This could be due to the fact that, men frequently stay out of their home late into the night and carry out more outdoor activities- like leaving their homes very early in the morning before it's dawn to go to their farms and often have to spend several days living in their farm houses, as such, exposing themselves to the mosquito vector. However, this was in contrast to previous findings, which showed a higher prevalence among woman than men in the mount Cameroon area, which is attributed to females spending more time outdoors at dusk and dawn than males to perform household chores, and as such, are more exposed to mosquito bites in this area [11,15].

Equally, malaria prevalence was significantly higher among the unemployed (p<0.001) and the GMPD decreased significantly with increase in income level (r = - 0.137, p = 0.002). This high prevalence among the unemployed is in concordance with the observation that poverty and socio-economic status of individuals can lead to increase in malaria prevalence [16,17] due to the fact that, the poorest people often have limited access to health services, particularly in rural settings.

The overall geometric mean parasite density (GMPD) of malaria among the people of Nkongho-mbeng in the current study was 6,869 parasites/μL of blood. This is similar to GMPD of 7,874 parasites/μL that was observed in a study carried out in Central Ghana [18] and 7,345 parasites/μL in Sudan [19]. The geometric mean parasite density of malaria was significantly associated with gender, age group, occupation and marital status. The differences observed in parasite density among different age groups with infants being the most heavily parasitized could be attributed to the fact that older people can control parasite densities than younger ones due to immunity build-up over time, from years of exposure to the parasite [20,21].

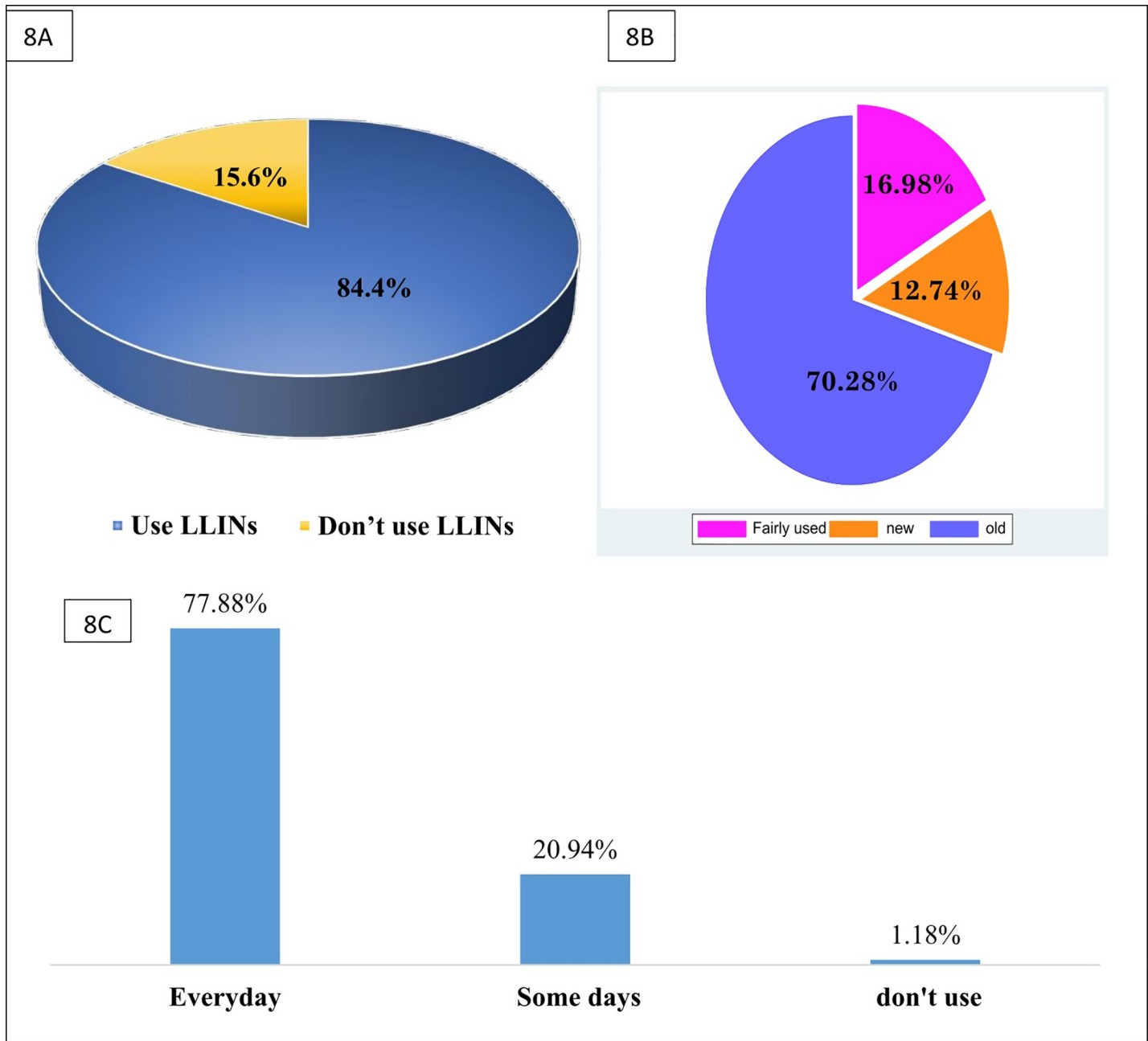

**Fig 8. A: Percentage of LLINs ownership in Nkongho-Mbeng, B: State of LLINs used in Nkongho-mbeng (Old = LLINs acquired more than 3year ago, fairly use = LLINs acquired between one and three years, and new = LLINs that were less than a year old) and C: Frequency of LLINs usage in Nkongho-mbeng.**

In relation to gender and marital status, the GMPD in this study was observed to be higher in males than females and in singles than in married people. This is contrary to what was observed in Muea, located in Fako division in the South West Region of Cameroon, where there was no significant difference in GMPD between sexes [22]. The parasite density in men could be due to the fact that, men spend more time outdoor and in farmhouses, and male children play outdoors in the evenings more often than female children and, are therefore, more exposed to mosquito bites. The fact that singles had higher parasite densities when compared

**Table 6. Prevalence and parasite density of malaria with respect to utilization of LLINs.**

| Characteristic | Category | Number examined | N° Positive (%) | GMPD (Parasite/μL) |
|---|---|---|---|---|
| **Presence of LLINs** | Yes | 422 | 53 (12.56) | 6481.89 |
| | No | 78 | 7 (8.97) | 10659.07 |
| **Significance** | | | **p = 0.371[a]** | **U = 15900.0, p = 0.399** |
| **Frequency of LLINs usage** | Everyday | 331 | 42 (12.69) | 6351.64 |
| | Some days | 89 | 12 (13.48) | 6007.99 |
| | don't use | 5 | 0 | 0.0 |
| **Significance** | | | **p = 0.678 [a]** | **H = 0.759, p = 0.684** |
| **State of LLINs** | Fairly use | 72 | 5 (6.94) | 14,236.42 |
| | New | 54 | 9 (16.67) | 10504.71 |
| | Old | 298 | 40 (13.42) | 5042.693 |
| **Significance** | | | **p = 0.218[a]** | **H = 2.869, p = 0.238** |

a: *P* value computed from chi square test H = Kruskal-Wallis test U = Mann-Whitney test.

with married persons, irrespective of age, suggests that other factors may account for high parasite load in singles. Some studies suggest that, marriage could improve health outcomes in a variety of ways: it may improve economic well-being [23], health outcomes by enhancing access to health care or lowering stress. In addition, a spouse may play an important role in monitoring and encouraging healthy behaviors [24].

Duration of stay in Nkongho-mbeng was observed to be significantly associated with parasite density, with individuals who have lived for less than 15 years in this community having a

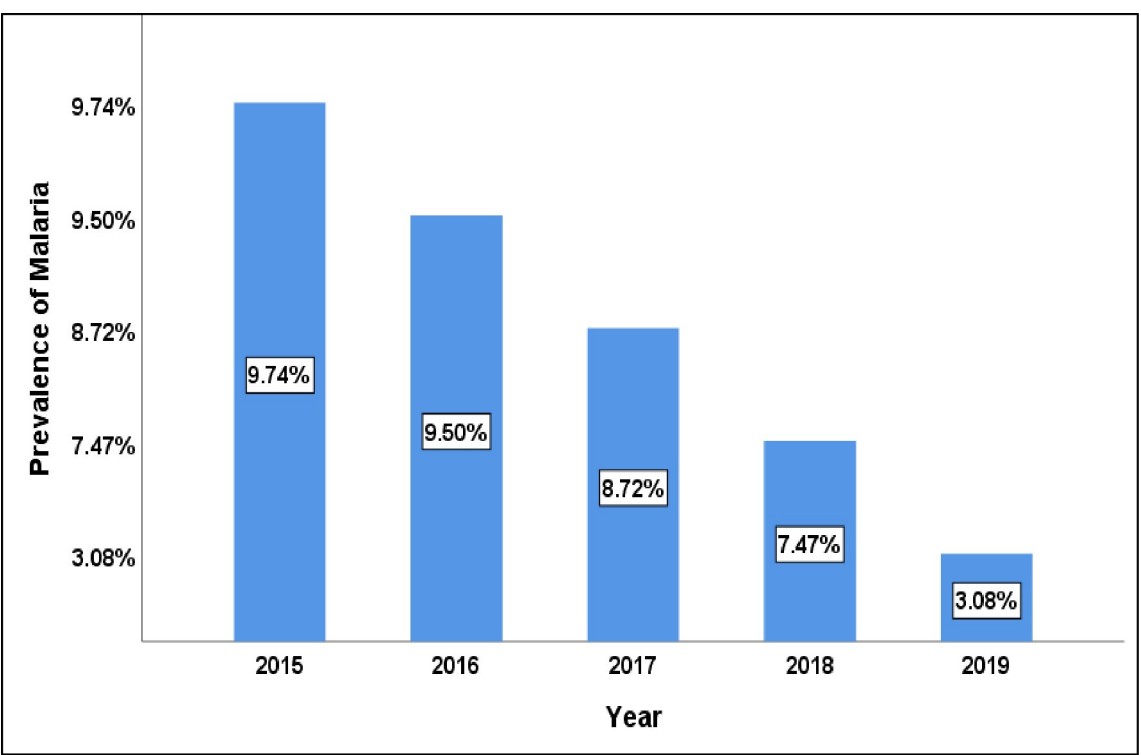

**Fig 9. Trends in Malaria prevalence per 1,000 inhabitants in Nkongho-mbeng from 2015–2019.**

higher parasite load than those who lived there beyond 15 years. A similar observation was made in Bolifamba [7]. This could be due to acquired immunity developed with prolonged duration of stay for people who have lived within this community longer than 15 years.

With reference to participants' income level, in this study, a correlation analysis showed that, the mean parasite density of malaria decreased significantly with increase in income level. This could be as a result of low nutritional balance and inability to access quality health care due to poverty in individuals with low income level. Our findings concur with observations elsewhere in sub-Saharan Africa, where study participants with low socio-economic status and income level were more liable to be infected than those with high income level [25,26].

The prevalence and geometric mean parasite density of malaria was seen to be significantly associated with presence of bushes and presence of stagnant water around houses. Participants living in bushy environments had significantly higher malaria prevalence and GMPD than those living in non-bushy areas. In the same line, those residing around stagnant water had significantly higher malaria prevalence (15.35%), but a lower GMPD than those residing in non-stagnant water areas. Malaria prevalence and GMPD were also higher among those living around streams, compared to those who were not living around streams, though not statistically significant. This could be from constant exposure to the vector as a result of living around vector breeding sites. This finding is in accordance with the findings in the high lands of Western Kenya, where participants that lived around streams had a higher GMPD compared to those that did not live around streams [27].

LLINs have been shown to reduce the burden of malaria infections, but coverage continues to be moderate in many parts of sub-Saharan Africa. However, our study showed a high ownership (84.4%) and usage (77.88%) by all persons in the area after the campaign, as reported elsewhere [28–30]. This high coverage of LLINs in Nkongho-mbeng could be due to the presence of more than two household members in almost all households in Nkongho-mbeng. Previous studies have shown that household whose family sizes are >2 have more chances of owning and using LLINs compared to their counterparts living in household with family size ≤ 2 [31,32]. This high LLINs coverage in Nkongho-mbeng could also be as a result of proper distribution of LLNs by the health facility during the campaigns. These findings are however in contrast what was observed in Fako division on the slope of Mount Cameroon, where there was a low ownership and usage of LLINs in rural areas compared to semi-urban settings [6].

The present study investigated the potential risk factors associated with malaria infection in Nkongho-mbeng. Bivariate and multivariate logistic regression analysis revealed that, residing around bushy areas and areas of stagnant water, and male participants were significantly associated with the prevalence of malaria infection among people of Nkongho-mbeng. These findings are consistent with those of other studies conducted in Bomaka and Molyko, located on the slope of mount Cameroon, and with a study on individual and housing factors influencing the incidence of malaria in Ethiopia, where residing around bushy areas and areas of stagnant water were identified as risk factors for malaria [15,33]. Living in Lekwe and Mbetta were also shown to increase the odds of being infected with malaria. Our study showed that, the prevalence of malaria in Lekwe (12.73%) and Mbetta, (14.29) was observed to be far higher than the 1.59% seen in Dinte. This finding could be due to the fact that the topography of Dinte is hilly compared to that of Lekwe and Mbetta which are more of lowland areas which harbor several breeding grounds for the malaria vector. Other studies have shown that, living around malaria vectors breeding grounds are a potential risk factor for malaria infection [15,34]. Multivariate logistic regression analysis also revealed that being a pupil was a possible determinant of malaria infection in Nkongho-mbeng. Similar findings have been reported elsewhere [35,36]. This could be as a result of pupils constantly playing around the malaria vector breeding site,

hence constantly exposed to the vector. This could also be as a result of a weak immune system in younger aged children.

The prevalence of malaria amongst the health care seeing population in Nkongho-mbeng decreased significantly from 2015 to 2019.This significant decrease in malaria morbidity could be due to constant emigration of people from Nkongho-mbeng to other regions where their children can have access to quality education and health care, due to the crisis in the English speaking regions of Cameroon. This could also be as a result of several inhabitants not being able to afford the cost of health care services due to economic hardship and poverty in Nkongho-Mbeng that is being aggravated by the crisis. As a result, many people may be seeking treatment elsewhere other than the health facility, as such are not being considered in the data that is sent to the South West Regional Delegation of Public Health from heath facilities.

In conclusion, the prevalence of malaria in Nkongho-mbeng located in the Nguti division of the South West Region of Cameroon is decreasing, with a currently low prevalence of 12%. Risk factors for the disease include; presence of bushes around homes and being unemployed, although there is a high level of ownership and usage of LLINs in the communities.

## Supporting information

**S1 File.**
(PDF)

## Acknowledgments

The authors are grateful to the people of Nkongho-mbeng for their participation in this study, and to St Theresa Catholic Health Center Mbetta, where the slides were observed for malaria parasites. Special thanks to Mr. Dobgima Ndanji Ndasi Mofor, for drawing the map of Nkongho-mbeng.

## Author Contributions

**Conceptualization:** Raymond Babila Nyasa.

**Data curation:** Raymond Babila Nyasa, Esendege Luke Fotabe.

**Formal analysis:** Raymond Babila Nyasa, Esendege Luke Fotabe, Roland N. Ndip.

**Supervision:** Raymond Babila Nyasa, Roland N. Ndip.

**Validation:** Raymond Babila Nyasa.

**Writing – original draft:** Raymond Babila Nyasa.

**Writing – review & editing:** Raymond Babila Nyasa, Esendege Luke Fotabe, Roland N. Ndip.

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
