## [Decision Letter · Decision Letter 0]

19 Mar 2021

PONE-D-21-00225

Trends in malaria prevalence and risk factors associated with the disease in Nkongho-Mbeng; a typical rural setting in the equatorial rainforest of the South West Region of Cameroon

PLOS ONE

Dear Dr. NYASA,

After careful consideration, we feel that your manuscript will likely be suitable for publication if the authors revise it to address critical points raised by the reviewer.  According to reviewer, there are some specific areas where further improvements would be of substantial benefit to the readers.   A major concern was related to the multicollinearity in the  regression analyses.  For your guidance, a copy of the reviewer' comments was included below.  

We look forward to receiving your revised manuscript.

Kind regards,

Luzia Helena Carvalho, Ph.D.

Academic Editor

PLOS ONE

Journal Requirements:

4. Please include additional information regarding the survey or questionnaire used in the study and ensure that you have provided sufficient details that others could replicate the analyses. For instance, if you developed a questionnaire as part of this study and it is not under a copyright more restrictive than CC-BY, please include a copy, in both the original language and English, as Supporting Information.

5. Please ensure you have thoroughly discussed any potential limitations of this study within the Discussion section, including the potential impact of confounding factors.

Reviewers' comments:

Reviewer's Responses to Questions

**Comments to the Author**

1. Is the manuscript technically sound, and do the data support the conclusions?

Reviewer #1: Yes

2. Has the statistical analysis been performed appropriately and rigorously? 

Reviewer #1: Yes

3. Have the authors made all data underlying the findings in their manuscript fully available?

Reviewer #1: Yes

4. Is the manuscript presented in an intelligible fashion and written in standard English?

Reviewer #1: Yes

5. Review Comments to the Author

Reviewer #1: This is a well-done and important contribution to the malaria literature. Updating the epidemiological profile of malaria prevalence is important for prioritizing control and prevention activities, and this study provides results that are useful for informing prevention and control. I have a few minor items I would like the authors to address:

1. Including a map showing the study area would be helpful to the reader.

2. It would be useful for the authors to address the possibility of multicollinearity in the multiple regression analyses between income level and variables associated with income level (housing structure, window screens, etc). Was multicollinearity assessed, and is it something that needs to be addressed? This is the most important comment I have.

3. Consider eliminating the p-values in tables 1 and 2 for the regression results. The 95% confidence intervals are the most important to include, and p-values are not as meaningful.

4. I found the high infant prevalence to be interesting. The authors addressed the role of no built immunity as a reason for higher prevalence in this age group, which is an appropriate explanation. I am curious about the sleeping habits of infants in the study areas in relation to LLIN use- do infants typically sleep with a guardian? Is it known whether infants typically sleep under LLINs?

5. In figure 7, consider using the same type of figure to maintain consistency (either use all pie charts or all bar graphs)

6. PLOS authors have the option to publish the peer review history of their article (what does this mean?). If published, this will include your full peer review and any attached files.

Reviewer #1: No

---

## [Author Response · Author response to Decision Letter 0]

13 Apr 2021

Dear Editor,

 We appreciate the significant contributions of the reviewers, which has given a face-lift to the manuscript. We are very grateful to them. However, our opinion differs with theirs on some minor issues, which could still be amended based on your recommendation. Details of the corrections are as below.

1) Including a map showing the study area would be helpful to the reader.

A map of the study area has been drawn and included in the manuscript.

2) It would be useful for the authors to address the possibility of multicollinearity in the multiple regression analyses between income level and variables associated with income level (housing structure, window screens, etc). Was multicollinearity assessed, and is it something that needs to be addressed? This is the most important comment I have.

Multicollinearity analysis has been carried out and the results included in the manuscript. 

3) Consider eliminating the p-values in tables 1 and 2 for the regression results. The 95% confidence intervals are the most important to include, and p-values are not as meaningful.

Although the 95% confidence intervals are the most important, novices to statistics easily interpret p-values than confidence intervals. Thus, having both will enable us to effectively communicate to a wider audience.

4) I found the high infant prevalence to be interesting. The authors addressed the role of no built immunity as a reason for higher prevalence in this age group, which is an appropriate explanation. I am curious about the sleeping habits of infants in the study areas in relation to LLIN use- do infants typically sleep with a guardian? Is it known whether infants typically sleep under LLINs?

There is no statistically significant difference (p=0.06) in LLINs usage between this age group and other age groups. Many other studies elsewhere have reported higher prevalence of malaria in this age group than the general population.

5) In figure 7, consider using the same type of figure to maintain consistency (either use all pie charts or all bar graphs)

If we use all pie charts or all bar charts, the manuscript may become banal and too uniform to the viewer, to incite interest in reading onward.

---

## [Decision Letter · Decision Letter 1]

26 Apr 2021

Trends in malaria prevalence and risk factors associated with the disease in Nkongho-mbeng; a typical rural setting in the equatorial rainforest of the South West Region of Cameroon

PONE-D-21-00225R1

Dear Dr. NYASA,

We’re pleased to inform you that your manuscript has been judged scientifically suitable for publication and will be formally accepted for publication once it meets all outstanding technical requirements.

Kind regards,

Luzia Helena Carvalho, Ph.D.

Academic Editor

PLOS ONE

Additional Editor Comments (optional):

Reviewers' comments:

Reviewer's Responses to Questions

**Comments to the Author**

1. If the authors have adequately addressed your comments raised in a previous round of review and you feel that this manuscript is now acceptable for publication, you may indicate that here to bypass the “Comments to the Author” section, enter your conflict of interest statement in the “Confidential to Editor” section, and submit your "Accept" recommendation.

Reviewer #1: All comments have been addressed

2. Is the manuscript technically sound, and do the data support the conclusions?

Reviewer #1: Yes

3. Has the statistical analysis been performed appropriately and rigorously? 

Reviewer #1: Yes

4. Have the authors made all data underlying the findings in their manuscript fully available?

Reviewer #1: Yes

5. Is the manuscript presented in an intelligible fashion and written in standard English?

Reviewer #1: Yes

6. Review Comments to the Author

Reviewer #1: I commend the authors for addressing the reviewer comments, and I am thankful for the reminder that p-values can be more useful to a broader audience. This is an excellent perspective that I will keep in mind myself moving forward in my own work. Additionally, the map is wonderful! Great work to the authors.

7. PLOS authors have the option to publish the peer review history of their article (what does this mean?). If published, this will include your full peer review and any attached files.

Reviewer #1: No

---

## [Editor Report · Acceptance letter]

29 Apr 2021

PONE-D-21-00225R1 

Trends in malaria prevalence and risk factors associated with the disease in Nkongho-mbeng; a typical rural setting in the equatorial rainforest of the South West Region of Cameroon 

Dear Dr. Nyasa:

I'm pleased to inform you that your manuscript has been deemed suitable for publication in PLOS ONE. Congratulations! Your manuscript is now with our production department. 

Kind regards, 

on behalf of

Dr. Luzia Helena Carvalho 

Academic Editor

PLOS ONE